# Temporal clustering of precipitation for detection of potential landslides

Fabiola Banfi[1,✉], Emanuele Bevacqua[2,*], Pauline Rivoire[3,4,*], Sérgio C. Oliveira[5], Joaquim G. Pinto[6], Alexandre M. Ramos[6], and Carlo De Michele[1]

[1]Dept. of Civil and Environmental Engineering, Politecnico di Milano, Milan, Italy
[2]Department of Computational Hydrosystems, Helmholtz Centre for Environmental Research – UFZ, Leipzig, Germany
[3]Institute of Geography and Oeschger Centre for Climate Change Research, University of Bern, Switzerland
[4]Institute of Earth Surface Dynamics, Faculty of Geosciences and Environment, University of Lausanne, Switzerland
[5]Centre for Geographical Studies, Institute of Geography and Spatial Planning (IGOT), Associated Laboratory TERRA, University of Lisbon, Lisbon, Portugal
[6]Institute of Meteorology and Climate Research, Troposphere Research (IMK-TRO), Karlsruhe Institute of Technology (KIT), Karlsruhe, Germany
[*]These authors contributed equally to this work.
[✉]Correspondence to Fabiola Banfi (fabiola.banfi@polimi.it)

**Abstract.** Landslides are complex phenomena that cause important impacts in vulnerable areas, including the destruction of infrastructure, environmental damage, and loss of life. The occurrence of landslide events is often triggered by rainfall episodes, single and intense ones or multiple occurring in sequence, i.e. clustered in time. Landslide prediction is typically obtained via process-based or empirical thresholds. Here, we develop a new approach that uses information on the temporal clustering of rainfall to detect landslide events and compare it with the use of classical empirical rainfall thresholds. In addition, we evaluate the performances of the two approaches combined together as a case study in the region of Lisbon in Portugal. We consider a dataset that categorises landslides into shallow and deep events, and a review of empirical rainfall thresholds that makes a good benchmark for testing our novel method. We show that the new approach based on temporal clustering overall has a good power of detecting landslide events, but has a skill comparable with the classic rainfall threshold method. While there is no clear outperformance of one method, the novel clustering-based method has a higher sensitivity despite a lower precision than the threshold-based method. For all approaches, the potential detection is better for deep landslides than for shallow ones. The results of this study could help to improve the prediction of rainfall-triggered landslides.

## 1 Introduction

Landslides are complex phenomena modulated by several interacting factors, which often cause catastrophic consequences in susceptible areas (Herrera et al., 2018; Maraun et al., 2022). Factors playing a role in landslide occurrences can be classified into two main categories (Dai and Lee, 2002): a) static predisposing variables, like surface-related characteristics (e.g., soil type and slope, lithology, aspect, topography) and b) triggering (dynamic) variables, like rainfall. Static variables describe factors such as surface-related characteristics, which do not immediately cause the landslides, but shape the landslide occur-

rence probability given certain conditions, e.g. weather conditions. Triggering variables describe factors that actually cause the landslides, for example intense precipitation events. Landslides often occur due to multiple factors, i.e. the combination of static and triggering variables or the combination of multiple triggering variables. Such combinations falls in the category of compound events, defined as "the combination of multiple drivers and/or hazards that contributes to societal or environmental risk" (Seneviratne et al., 2012; Leonard et al., 2014; Zscheischler et al., 2020). Hence, compound events analyses can help detect and predict this natural hazard. For example, a compound event perspective allows for understanding the probability of landslides after wildfires, since wildfires change soil characteristics, thereby increasing landslide likelihood (Di Napoli et al., 2020). Notably, temporally compounding precipitation events, i.e. temporal clustering of moderate to extreme rainfall events, can trigger landslides by rising groundwater levels of deep soil and rock layers (Bevacqua et al., 2021).

The latter example is relevant since one of the most important drivers of landslides is rainfall, either short and high-intensity episodes or long-lasting ones (Van Asch et al., 1999). Deep landslides, characterised by a slip surface deeper than about 1.5 m, are usually initiated by multiple moderate-intensity storms, occurring over weeks or months (Trigo et al., 2005). Such wet periods can result in high soil moisture and pore water pressure, necessary to trigger deep movements (Chen et al., 2017). In contrast, shallow landslides and debris flow take place under a broader range of rainfall conditions, and they are more often associated with short-duration and high-intensity rainfall events (Corominas and Moya, 1999). In line with the above, for the North of the Lisbon region, Bevacqua et al. (2021) showed about 70%–83% of deep landslides were preceded by a temporal cluster of precipitation events (over 23–90 days before the event). In contrast, only 7%–9% of shallow landslides were preceded by a cluster of precipitation (over 4–25 days before the event).

In addition, recent works have considered the combined role of antecedent and peak precipitation for in the initiation of landslides (Kim et al., 2021; Nocentini et al., 2024). Antecedent rainfall, inducing a gradual increase in soil moisture and groundwater level, is an important factor in the determination of the slope stability and, therefore, the initiation of landslides (Rahardjo et al., 2001; Rahimi et al., 2011; Lee et al., 2012). Deep landslides, for example, may often be associated with monthly to seasonal fluctuations of the groundwater table. When the water table is high, light to moderate rainfall may provide sufficient water to trigger slope movement (Christopher M. Fuhrmann and Band, 2008).

Despite the advances in our comprehension of the major drivers of landslides (Tehrani et al., 2022), processes linking rainfall and landslide occurrence are not yet fully understood, thus modeling the occurrence of this natural hazard is not straightforward (Guzzetti et al., 2007; Tehrani et al., 2022). The prediction of landslides is mainly approached by defining rainfall thresholds that separate critical and non-critical rainfall events, i.e. events that are more or less likely to trigger landslides (Guzzetti et al., 2007; Segoni et al., 2018). Thresholds for landslide occurrence can be classified in two main typologies: (i) process-based and (ii) empirical. (i) Process-based thresholds are derived from slope stability models and allow for deriving the precipitation amount necessary to trigger the landslide, its date and location. However, obtaining such thresholds is often difficult due to data limitations. (ii) Empirical thresholds are estimated by studying past landslides. They are mainly obtained (a) from precipitation measurements during specific rainfall events, causing (or not) landslides, or (b) from antecedent conditions, i.e. the total precipitation preceding a landslide over different durations. Additional typologies of thresholds were proposed in the literature and we refer to Guzzetti et al. (2007) for a complete overview. An innovative rainfall threshold is the one by Nocentini

et al. (2024), which instead of considering thresholds based either on antecedent rainfall or peak intensity rainfall, propose a 3D threshold implementing both conditions.

The main drawback of empirical rainfall thresholds is that they are calibrated for a specific region, therefore implicitly including geomorphological and meteorological characteristics of the specific site. This prevents having a threshold that can be readily applied to different regions (Abbate et al., 2021). To overcome this limitation, global empirical rainfall thresholds have been developed, however they result in lower performances, with a high rate of false alarms (Guzzetti et al., 2007). Hence, a proper general rule is still missing (Zêzere et al., 2008).

Building on the association of temporal clustering of rainfall events and landslides identified by Bevacqua et al. (2021), we investigate whether a compound event perspective may benefit the estimation of the probability of occurrence of rainfall-triggered landslides, both shallow and deep ones. To this end, we develop a new approach that uses information on the temporal clustering of rainfall to detect landslides, and we compare it with the use of classical empirical rainfall thresholds. In addition, we evaluate the performances of the two approaches combined together. The newly proposed method is an empirically based approach, i.e. does not require any data beyond precipitation and landslide occurrence data. While it does not require a regression fitting, it still requires the definition of a quantile-based threshold of precipitation to identify critical precipitation events. The dependence on the quantile is however investigated to assess whether it may be more easily exported to other sites with respect to regression based thresholds.

We apply our approach to two landslide data sets, in the region of Lisbon in Portugal. For the Lisbon region, Zêzere et al. (2015) provide a collection of landslide events, subdivided into shallow and deep events, and a review of empirical rainfall thresholds, making it a good benchmark to test the novel method. An interesting characteristic of the two datasets is that they deal with areal and not single landslides. Older datasets, like the AVI dataset (Guzzetti et al., 1994), an inventory of landslides and floods that occurred in Italy, record landslides triggered by the same precipitation system as separate events. This is an important limitation when interested in identifying the landslides' triggers, therefore the subdivision between single landslide events and multiple landslide events is gaining consensus recently. For example, this is the case of the new catalog Frane Italia by Calvello and Pecoraro (2018), the LAND-deFeND database (Napolitano et al., 2018) or the work of Crozier (2017).

The paper is organized as follows: in Sec. 2 we discuss the data and the methodology, results and discussions are provided in Sec. 3 and 4, while conclusions are given in Sec. 5.

## 2 Methodology

Here, we assess the ability of temporal clustering of rainfall in estimate the probability of landslides occurrence, and we compare it with the use of classical rainfall thresholds. This was done for two landslides data sets in the Lisbon region (described in Sec. 2.1) in four steps: i) rainfall thresholds are estimated (Sec. 2.2), ii) the presence of temporal clustering of rainfall before each landslide event and before each day is studied (Sec. 2.3), iii) alarm and non alarm periods, with the selected approaches, are identified (Sec. 2.5), and iv) the performance of the different approaches in landslide detection are assessed with the help of different metrics (Sec. 2.5).

## 2.1 Landslides and precipitation data

High levels of destruction caused by natural disasters of hydro-geomorphologic origin had been recorded in Portugal since the late XIX century (the DISASTER database, Zêzere et al., 2014). Until 2015, 281 disastrous landslide records registered considerable adverse consequences in mainland Portugal, such as, loss of life (273 deaths) or injury, displaced people ($> 1600$), property damage, economic disruption, or environmental degradation (Pereira et al., 2020). In this context, the Portuguese Western Meso-Cenozoic border, in which the north of Lisbon/Lisbon region is included, is recognized as one of the most high-risk areas (Fig. 1). In this region the number of deaths and missing people represents 21% of the total in the country, but the displaced people due to landslides reaches almost 70%, mainly associated to slow moving deep-seated rotational and translational slides (Zêzere et al., 2014). Regionally, landslides are primarily controlled by lithology, geological structure, hydrogeological conditions and slope. In the last, shallow soil slips tend to concentrate on slopes steeper than $15°$ and deep-seated landslides on gentle to moderate slopes ($5° - 15°$), which are more favourable to water infiltration and storage (de Brum Ferreira and Zêzere, 1997).

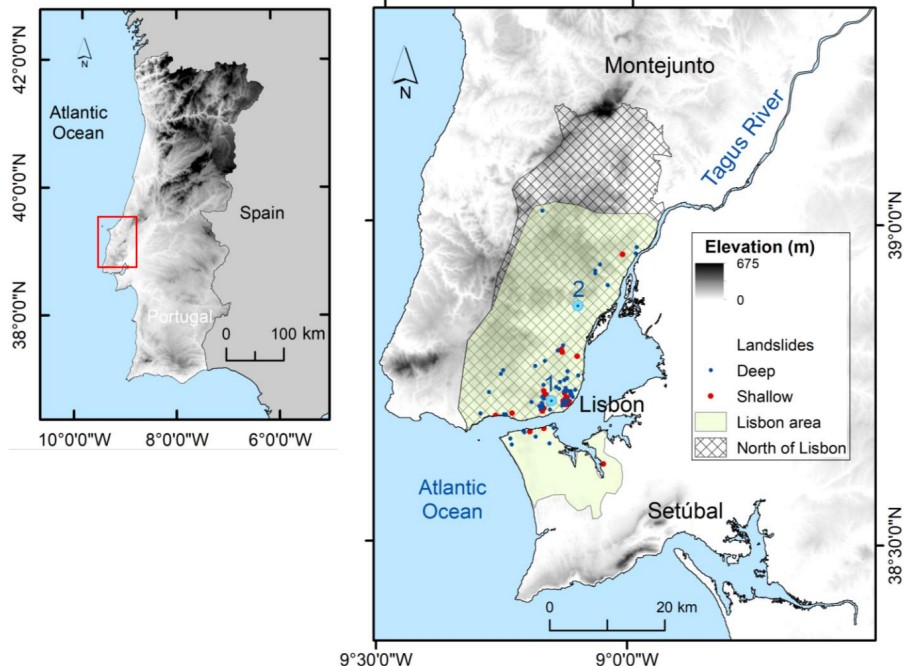

**Figure 1.** North of Lisbon and Lisbon study areas. Dots represent the landslide cases associated to the different landslide events described in the Disaster Database available at Zêzere et al. (2022). Red dots represent shallow landslide events and blue dots the deep landslide events. Rainfall gauges (blue circles): 1) Lisboa Geofísico; 2) São Julião do Tojal. The landslides associated to the north of Lisbon dataset were not included to improve readability.

We selected the events from two data sets of landslides in the Lisbon region by Zêzere et al. (2015), since they were characterized by both shallow and deep events (Fig. 1). The first data set covers the area of Lisbon, from 1865 until 2010, and it includes 39 events, which were collected from newspapers (Vaz et al., 2013). For this inventory, the landslide type is estimated based on the associated critical pair of rainfall intensity and duration preceding the landslide. The second data set covers the North of Lisbon region, from 1956 until 2010, and it includes 25 events (Zêzere and Trigo, 2011). Data were obtained from multiple sources: technical and scientific documents, fieldwork, and interviews with the local population. The distinction between deep and shallow landslides is based on the slip surface depth, with lower than 1.5 m depths associated with shallow landslides. For additional information about the two data sets, please refer to Zêzere and Trigo (2011); Vaz et al. (2013); Zêzere et al. (2015). It is important to highlight that we are working here with landslide event dates and not with individual landslides, i.e. multiple landslides can occur during a landslide event (See Sec. 4.3).

Daily precipitation time series for the Lisbon area during 1863-2018 was obtained from the meteorological station of Lisboa-Geofísico and it can be downloaded from the Portuguese Institute for Sea and Atmosphere (IPMA). To study the North of Lisbon region, the daily IBERIA02 data set was employed (Belo-Pereira et al., 2011). The data set has a $0.2°$ resolution (approx. 19.6 km) and the grid cell closest to the landslide events was chosen.

After an examination of the yearly distribution of landslide events in both data sets, we narrowed the analysis to the November–March period, when the majority of landslides occurred.

## 2.2 Empirical rainfall-thresholds estimation

The new method proposed in this work was compared with the empirical rainfall thresholds. One of the critical steps in the evaluation of rainfall thresholds is the definition of the critical rainfall period associated with each landslide event. We follow the approach of Zêzere et al. (2015), who assess the critical pair duration-quantity of rainfall preceding the landslide event as the one associated with the highest return period. The method works as follows: (i) the precipitation total along the whole time series for each time window of size from 1 to 90 days is calculated, (ii) for each duration the maximum over each year is computed, (iii) the series of maxima is fitted using a Gumbel distribution, (iv) the precipitation total preceding the landslide events, for windows of 1 to 90 days ending the day of the event, is computed (see Fig. 2), (v) the return period of each of these duration-quantity pairs is evaluated using the parameters of the Gumbel distribution estimated in step (iii), (vi) for each landslide event, the duration-quantity pair with the highest return period is considered as the critical rainfall return period. Once the critical duration-quantity pair for each event is collected, a regression is fitted to this data to obtain the critical rainfall for each possible duration. For North of Lisbon region they fitted a linear regression, while for the Lisbon area they fitted a potential regression, a lower limit linear threshold that limits rainfall conditions below which no landslides occurred in the record, and an upper limit potential threshold above which landslides have been systematically observed. This regression curves are the rainfall threshold for landslide initiation.

In this work, we computed the empirical thresholds for the two datasets, following the same procedure of Zêzere et al. (2015). We selected the potential regression for the Lisbon area and the linear regression for the North of Lisbon region, based on the results of Zêzere et al. (2015), for comparison with the proposed method based on precipitation clustering.

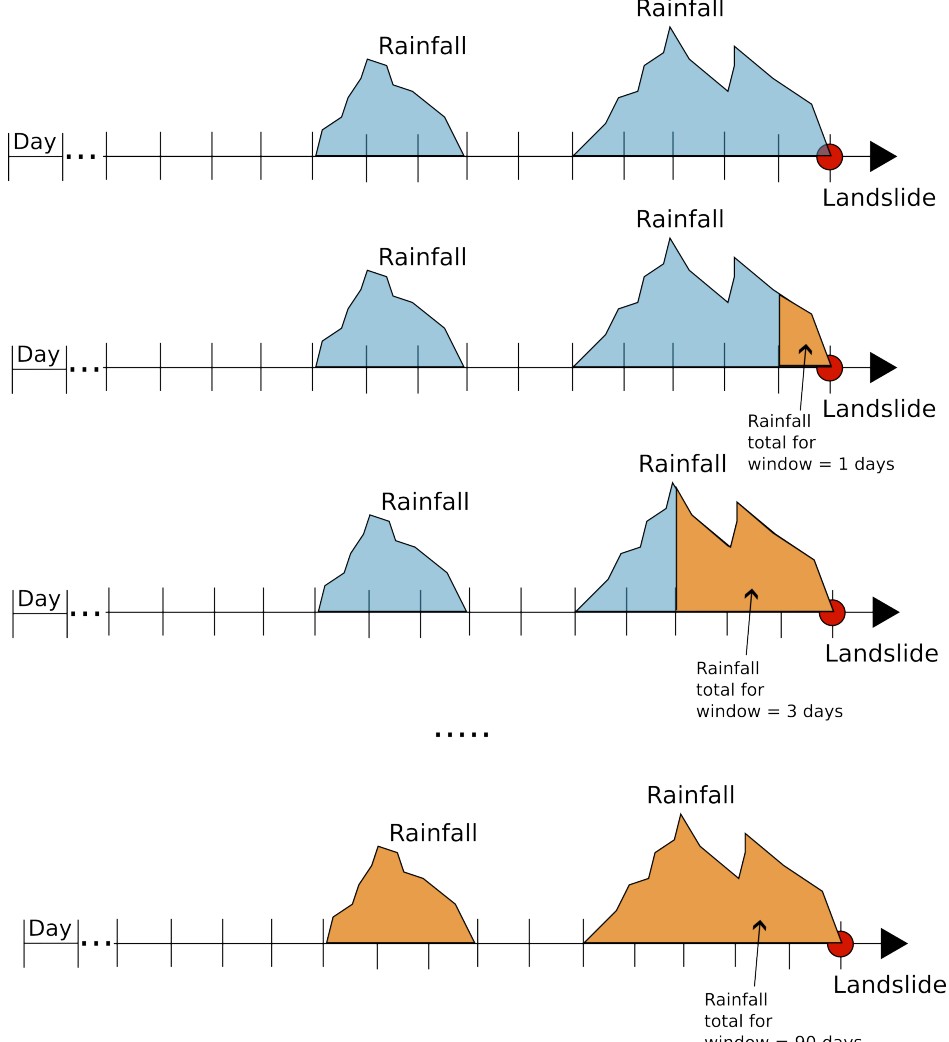

**Figure 2.** Illustration of step iv (presented in section 2.2) for the computation of rainfall thresholds. The first raw represents the time series of precipitation prior to the occurrence of the landslide (shown as a red dot). The following raws show the accumulated precipitation (in orange) over time windows of different lengths prior to the landslide. By expanding the time window, more precipitation before the landslide is considered.

## 2.3    Identification of temporal clustering of rainfall

In order to evaluate the occurrence of temporal clustering of rainfall before each landslide event, we employed the approach proposed by Banfi and De Michele (2022), extending the method of Bevacqua et al. (2021), with a different consideration of the effective window size as discussed below. The method is based on the idea that, once high-frequency clustering is removed,

the number of precipitation events in a window is distributed as a Binomial distribution, in the absence of lower frequency clusters.

The Binomial distribution is the discrete probability distribution of the number of successes in n independent trials. Each trial can have only two outcomes, yes or no, and the probability of having a yes in each individual trial is equal to p. The parameters of the distribution are therefore p and n. In this case each day is a trial and the outcome for each day is wet (yes) or dry (no). The probability of having a yes or no in a day is independent from the same probability in the other days, as an example, if day i is wet, it is not more probable that the following days are wet. This is true if the precipitation events are not clustered. If they are clustered, then, if day i is wet, it is more probable that the following days are wet. The assumptions of the binomial distribution are therefore not respected.

This method requires a series of independent precipitation events (i.e. stemming from different perturbations). To identify the events, a threshold $u$ of daily precipitation is used. To assure independence between them we adopt a (high-frequency) declustering procedure (Coles, 2001): when two events are separated by less than $r$ days (see Ferro and Segers (2003) for its estimation), only the first event is retained. Then, the probability of exceedance above $u$, $p$, is computed as $\frac{N}{L-D}$ where $N$ is the total number of independent events, $L$ the total length of the series and $D$ is the number of days in which events were removed with high-frequency declustering.

Let $x$ be the day for which the occurrence of a preceding temporal clustering of rainfall needs to be tested, $w$ a time window ending the day $x$, and $n$ the number of exceedances inside $w$. A statistical test for the presence of cluster can be defined (Fig. 3): the null hypothesis $H_0$ is defined as the absence of temporal clustering of precipitation inside $w$ and the alternative hypothesis $H_1$ as the presence of temporal clustering of precipitation inside $w$. $H_0$ is rejected if $n$ is larger than what is expected from a Binomial distribution of parameters $w_{\text{eff}}$ and $p$, with a $\alpha\%$ significance level. Here, $w_{\text{eff}}$ is the effective window that takes into account the effect of the (high-frequency) declustering on the low-frequency clustering, equal to $w - d$ where $d$ is the number of days exceeding $u$ in $w$, but already proceeded by at least one day exceeding $u$, and at most $r$.

In the declustering procedure explained in Barton et al. (2016), we set $r$ to 2 days, and we tested different values for $u$: namely, the 70th, 75th, 80th, 85th, 90th, and 95th percentile of above zero daily precipitation. To compute $u$, we considered only the November - March period, during which the selected landslide events occurred. Regarding the time window $w$, we tested window sizes varying from 4 to 90 days (Bevacqua et al., 2021). The window size is chosen to include the temporal scales affecting both shallow and deep landslides. Windows shorter than 4 days were not considered since, after high-frequency declustering, more than 3 days are required to have at least two precipitation events. Given that the maximum window size is 90 days, the probability $p$ was computed considering the period August-March.

To obtain the region of significance of the test, we must take into account both the discreteness of the p-value and the presence of multiple dependent tests. We therefore adopted the Benjamini-Hochberg procedure (Benjamini and Hochberg, 1995) applied on midP-values (Heller and Gur, 2011), with a significance level of 5%. The procedure is the following: (i) the p-values $p_i = P(X \geq x_i)$ and $p_{i_-} = P(X \geq x_{i+1})$ are computed, (ii) the midP-value is obtained summing $p_i$ and $p_{i_-}$, (iii) the critical value for each individual midP-value is obtained as $i/m \times \alpha$ where $i$ is the rank of the p-value and $m$ the number of tests, and (iv) the overall critical p-value is obtained as the largest individual midP-value smaller than the critical one. The

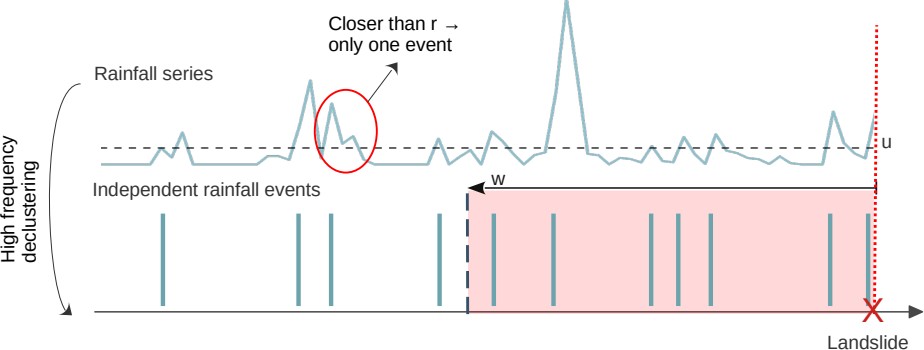

**Figure 3.** Illustration of the detection method adopted for temporal clustering identification Bevacqua et al. (2021); Banfi and De Michele (2022). The top curve represent the rainfall time series, the peaks on the second line represent the selected heavy rainfall events, after high-frequency declustering. $r$ is the minimum number of days between two peaks. $u$ is the threshold for heavy rainfall. $w$ represents the time window (in days, red shading) before the occurrence of a landslide (red cross). We count $n$ the number of events in the window $w$

correction was applied separately on each day with temporal clustering and considering together all the windows from 4 to 90 days.

This procedure was applied with two different goals: a) investigating the atmospheric drivers of the clustered precipitation events triggering landslides (see Sec 2.4) and b) testing the added value of the temporal clustering in landslides detection (see Sec 2.5).

## 2.4 Connection between landslides

If two landslide events are triggered by the same precipitation clustering event, the second landslide event does not bring additional information about the triggering conditions. We therefore investigated the connection between multiple landslide events that occurred during the same season (Fig. 4). First, we selected seasons with more than one landslide event associated with precipitation clustering. Then, for each season, i) we removed all the precipitation events preceding the first landslide event, ii) we computed the presence of rainfall clustering preceding the second event with the modified series, iii) we removed all the precipitation events preceding the second landslide event, iv) we computed the presence of rainfall clustering preceding the third event with the modified series, and so on for all the events in the same season. If two (or more) consecutive events are associated with temporal clustering of precipitation in the original series, while with the modified series they are not, then the two (or more) consecutive events are considered connected (Fig. 4).

## 2.5 Landslides detection

We considered three approaches to estimate the probability of occurrence of landslides: a) the presence of temporal clustering of rainfall, b) a rainfall-duration pair above the rainfall threshold and c) the AND combination of a) and b). The approach a) was tested for different thresholds $u$ (Sec 2.3). The approach b) was tested with the rainfall thresholds reported in Sec 2.2.

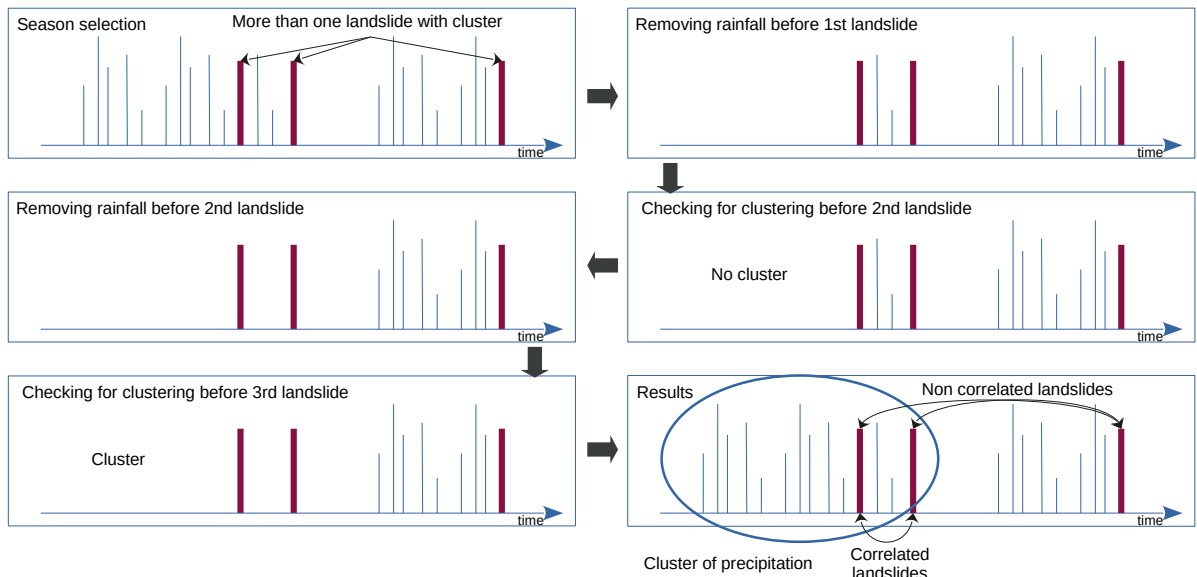

**Figure 4.** Workflow to determine whether the triggering conditions of two landslide events (or more) are related. We gradually remove the precipitation events prior to each landslide (in chronological order) and check whether it removed the presence of temporal clustering of precipitation triggering the following landslide.

To evaluate the performance of the methods, we separated the dataset into a training and validation set. Due to the limited length of the sample we used a leave-one-out cross-validation. This procedure consists of performing a number of experiments equal to the number of seasons in the record, and for each experiment one year is used for testing and all the remaining ones for training.

In this way we had a number of regression parameters equal to the number of years (Fig. 5). For the method based on temporal clustering we have different threshold values for each year since the value of the quantile would be different.

Then we computed three metrics to summarize the results, reported in Tab. 1. These metrics are based on four quantities: a) the true positive (TP), which is the number of events accurately predicted, b) the true negative (TN), which is the number of accurate predictions of the absence of an event, c) the false positive (FP) which is the number of predicted events that did not occur, d) and the false negative (FN) which is the number of non-predicted events. The three metrics are: 1) the sensitivity (also called probability of detection) which is the percentage of events that are forecasted, 2) precision (also called success ratio) which is the ratio of hits to the total number of events forecasted, and 3) the critical success index (CSI) which is the ratio of the number of hits to the total number of forecasts that were made or needed, and can be computed from the previous two measures. Two components therefore contribute to the CSI: one that depends on TP and FN thus highlighting the methods that have a high hit rate and miss few landslides, and the other one on TP and FP, thus highlighting the methods that have a high hit rate and produce few false alarm.

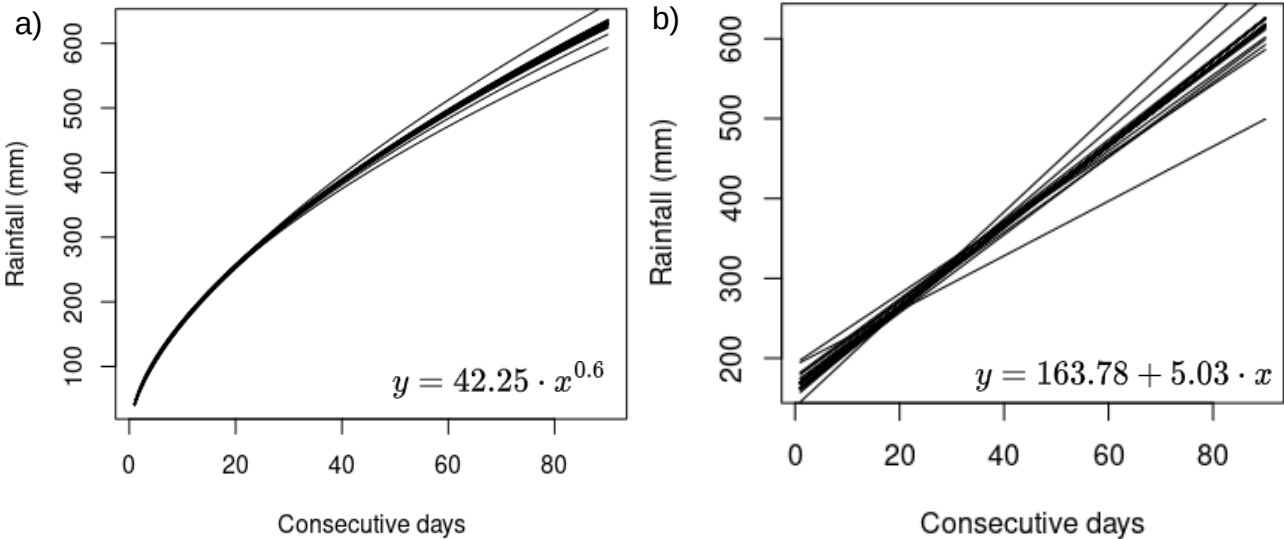

**Figure 5.** Empirical thresholds obtained from the calibration procedure. Panel a) Lisbon area, Panel b) North of Lisbon region. In the bottom right is reported the equation of the regression curve with the average parameters.

The sensitivity, the precision and the CSI highlight and summarize different aspects of the methods' performance. We did not consider true negative in our metrics because we wanted to focus on extreme events detection. There are many "0" events but predicting the absence of events is not central here. We want to focus on landslides, their occurrence, how often we caught them or had a false alarm.

| Sensitivity (POD) | Precision (SR) | Critical Success Index (CSI) | |
|:---:|:---:|:---:|:---:|
| $\dfrac{TP}{TP+FN}$ | $\dfrac{TP}{TP+FP}$ | $\dfrac{TP}{TP+FN+FP}$ or | $\dfrac{1}{\text{POD}^{-1}+\text{SR}^{-1}-1}$ |

**Table 1.** Performance metrics of predictive methods. The acronyms TP, FP, and FN stand respectively for true positive, false positive, and false negative.

210

For each day and for each method, we checked whether the method detected a potential landslide event. To compute true positives, we checked how many times the potential landslide detected occurred in the day of a real landslide event or on one of the 7 days preceding it. False positives were computed as the number of November-March periods without landslides during which at least one potential landslide event was erroneously detected.

 ## 3 Results

### 3.1 Temporal clustering of rainfall

The relation between the presence of multiple precipitation events in succession and the occurrence of landslide events is explored in the following, with a particular distinction between deep and shallow landslide types. In line with the results of Bevacqua et al. (2021), we observed a different behaviour between deep and shallow landslides in both data sets (Fig. 6). In

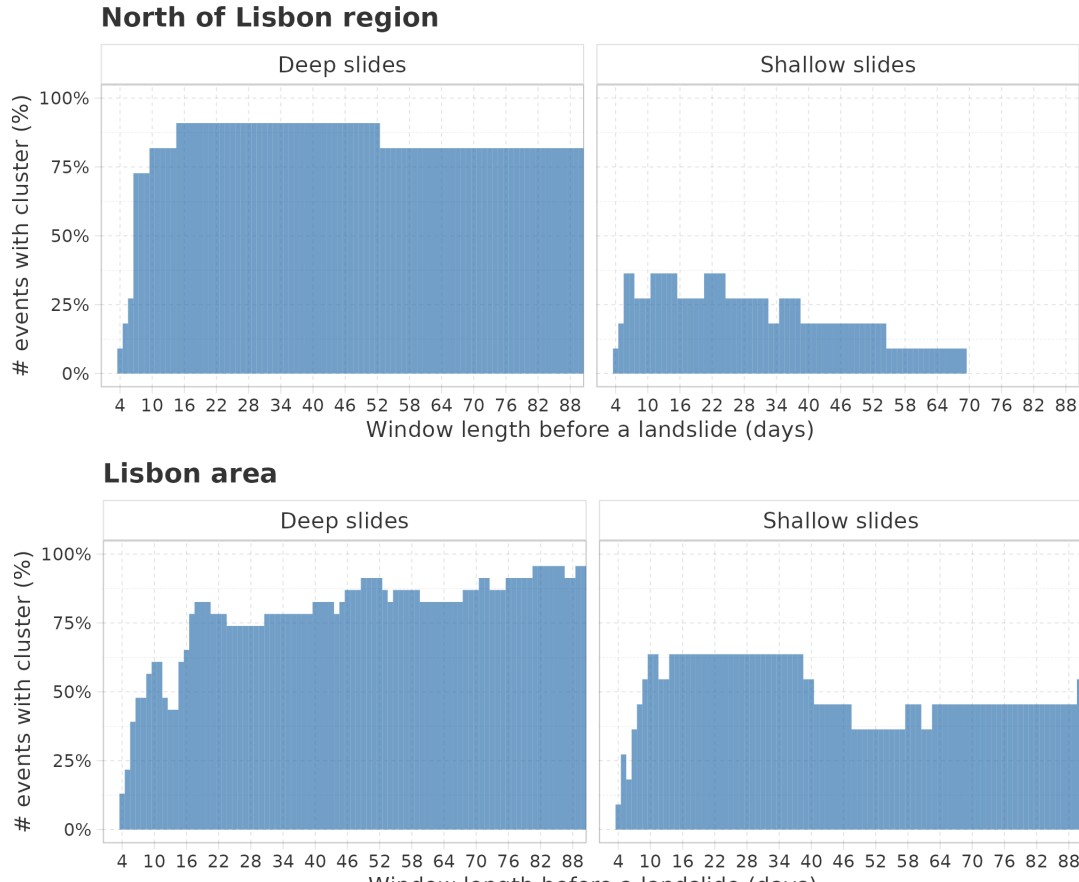

**Figure 6.** Percentage of windows with temporal clustering of rainfall preceding landslide events, for the two data sets, separated between shallow and deep landslides. Results are reported for $u$ equal to the 80th quantile of daily positive precipitation.

 particular, shallow landslides are associated with a lower percentage of rainfall clustering, mainly relegated to shorter windows. In contrast, the presence of clusters preceding deep landslides is a recurrent feature, mainly when window sizes greater than 10-15 days are considered. For the North of Lisbon, considering precipitation events larger than the 80th quantile, we found a higher presence of clusters for deep landslides, compared to the Lisbon area, and a significantly lower presence of clusters for

shallow landslides. In this dataset, in fact, the presence of temporal clustering of rainfall in time periods longer than 70 days

was never observed before shallow landslides. In the North of Lisbon region, 91% of deep landslides (10 out of 11) and 36% of shallow landslides (4 out of 11) were associated with a cluster of rainfall in at least one window. In the Lisbon area, 95.6% of deep landslides (22 out of 23) and 63.6% of shallow landslides (7 out of 11) were associated with a rainfall cluster in at least one window.

## 3.2    Sequence of landslides

In both data sets, landslide events occurring in sequence, i.e. multiple landslides in the same season, were recorded (Fig. 7). Lisbon area has around 14% of seasons with at least one landslide and 4% with more than one, the same percentages for North of Lisbon region are 24.5% and 7%.

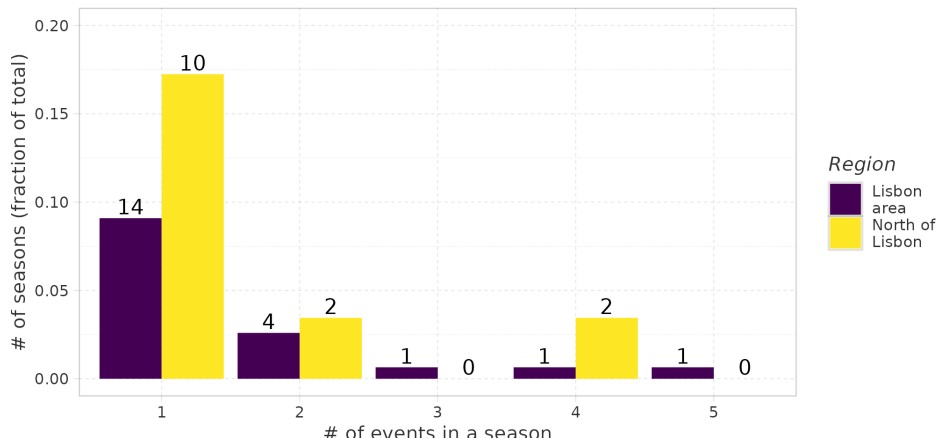

**Figure 7.** Fraction of season with 1, 2, 3, 4 or 5 landslides (y-axis) and absolute number (label above bars).

For the seasons with more than one landslide event we looked whether the events were connected, i.e. associated with the same cluster of precipitation, and we found that many of these multiple events are indeed connected (Figs. 8 and 9). In the

North of Lisbon region (Fig. 8), we found a sequence of connected deep landslides in the November-March seasons 1995–1996 and 2000–2001. In the period 1958–1959, two unconnected shallow landslide events were recorded of which only the first one is characterized by the presence of temporal clustering. In the period 1989–1990, instead, we observed a combination of deep and shallow landslide events, with and without a preceding temporal clustering of rainfall. Here, with the approach reported in Sec. 2.4, we associated the second and third landslide to the same rainfall cluster. The most frequent combination of multiple

events in the Lisbon area (Fig, 9) is the occurrence of multiple connected deep landslides (periods 1891-1892, 1958-1959, 1962-1963, and 1995-1996). One exception is the succession of two connected shallow landslides during the period 1937-1938. The two remaining periods (1946-1947 and 1968-1969) are characterized by the presence of both typologies, shallow and deep, with different combinations of cluster presence or absence. Also in these seasons, landslides were associated with the same rainfall cluster.

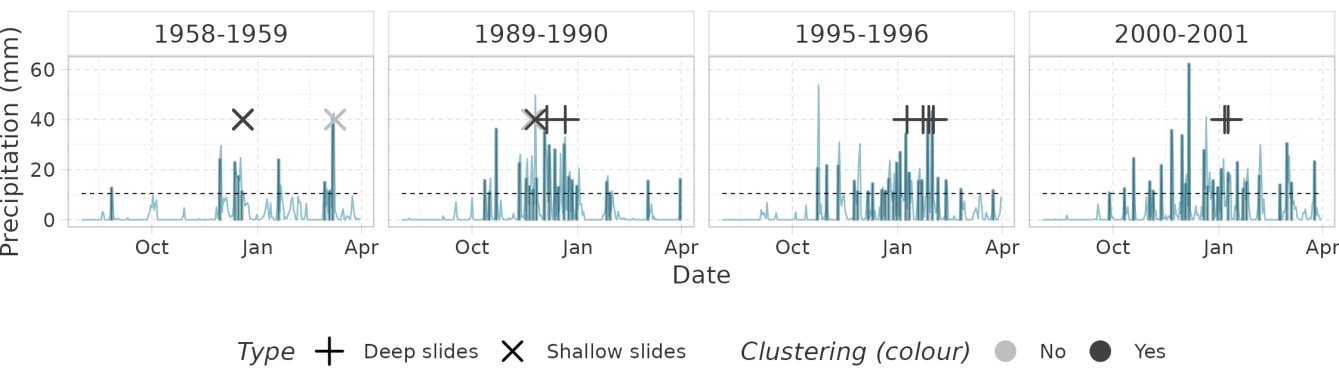

**Figure 8.** Precipitation time-series (light-blue line) and exceedances (blue bars) above the 80th quantile of wet days (dotted line) for North of Lisbon region, in some selected periods. Points corresponding with the occurrence of landslides are reported with crosses. The colour of the crosses indicates the presence of precipitation clustering prior to the landslide.

### 3.3   Landslides detection

The approach based solely on cluster occurrence is compared with the classical rainfall thresholds in Fig. 10. The approach based on the combination of cluster occurrence and rainfall thresholds is compared to the use of solely rainfall thresholds in Fig. 11. In both cases, we report the performance by the validation procedure.

In a nutshell, introducing the method based on temporal clustering of precipitation, and comparing it to the regression curve from Zêzere et al. (2015), we have better performances with the new method, for both sites and type of landslides in terms of sensitivity, for nearly all quantile levels. However, performances in terms of precision are fairly different depending on the site, with better performances using the method based on temporal clustering with the Lisbon area dataset and worse performances with the North of Lisbon dataset.

Combining the two indexes with CSI, we have lower performances with the proposed method for North of Lisbon region and deep landslides while higher performances for North of Lisbon region and shallow landslides and both landslide event types for the Lisbon area.

To make the best out of the cluster-based approach and rainfall-thresholds, we combine the precipitation clusters method with the rainfall thresholds one, and compare it with the sole use of rainfall thresholds (Fig. 11). To combine them, we consider to detect a landslide event when both methods detect an event. By construction, this hybrid method results in a number of TP and FP equal to or lower than the one of rainfall thresholds alone. In the north of Lisbon region, we obtain better or equal performances for all indices and all landslide types for nearly all choices of the quantile level, with a low influence of this last parameter. For deep landslides in the Lisbon area, we obtained a better performance with respect to rainfall thresholds, regarding both precision and CSI but a lower one for sensitivity.

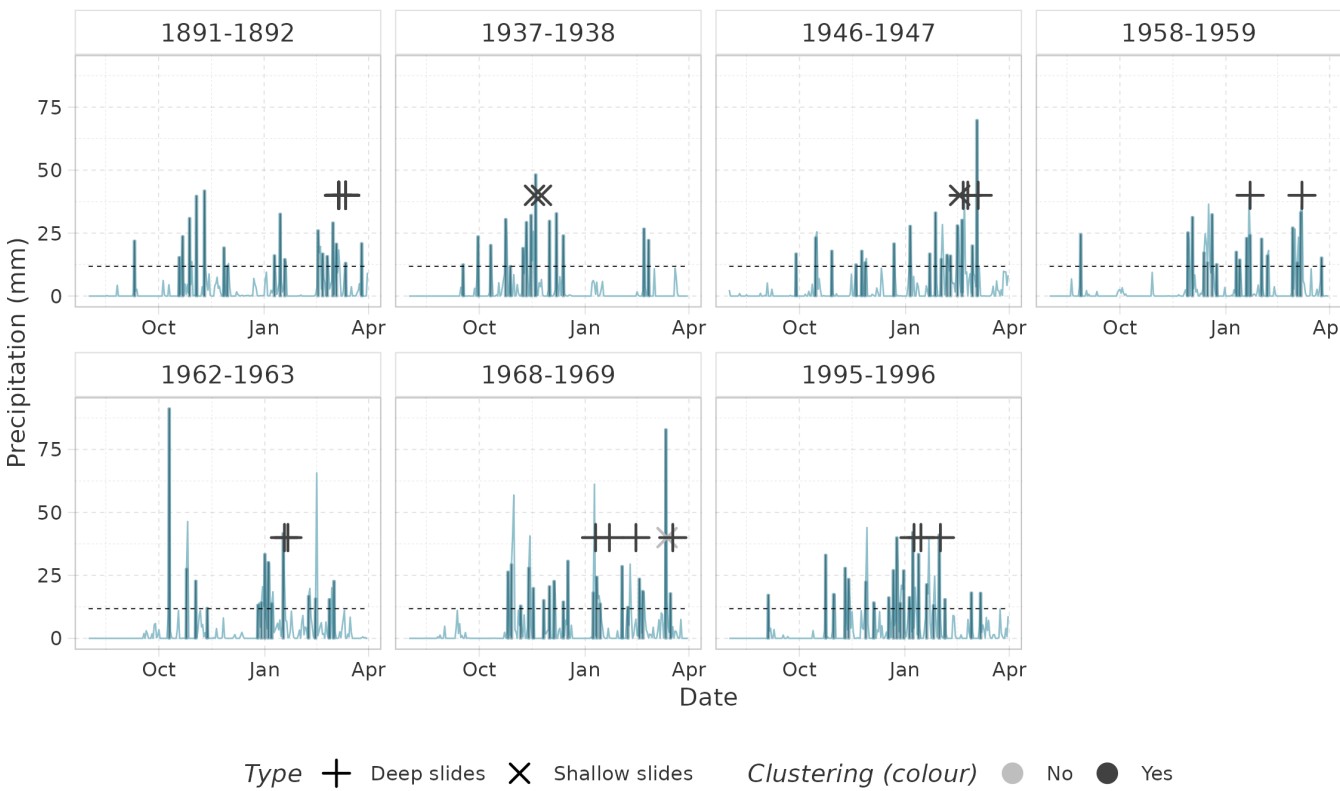

**Figure 9.** Same as Fig. 8 but for the Lisbon area.

## 4 Discussion

### 4.1 Association between landslides and clustering of rainfall

As already observed by Bevacqua et al. (2021), we found different characteristics of the precipitation events triggering deep and shallow landslides. Precipitation clustering was observed before more than 90% of deep landslides in the record. In particular, it was significant more frequently when longer time windows were considered On the contrary, clustering is present to a lesser extent in shallow landslides, and mainly on smaller windows.

Investigating the connections between subsequent landslide events is important to properly understand the associated risk. Here, we observed that multiple landslides, occurring close in time, can be almost always attributed to a common cluster of precipitation. Interestingly five landslides occurred during the March-November period 1968–1969 in the Lisbon area, three deep landslide events, a shallow one, and again a deep one. Here we can see the different mechanisms generating the two landslide event types; in fact, we found a cluster associated with all deep movements and no cluster associated with the shallow one that was instead preceded by an intense precipitation event. We can also notice that the period 1995-1996 was characterized

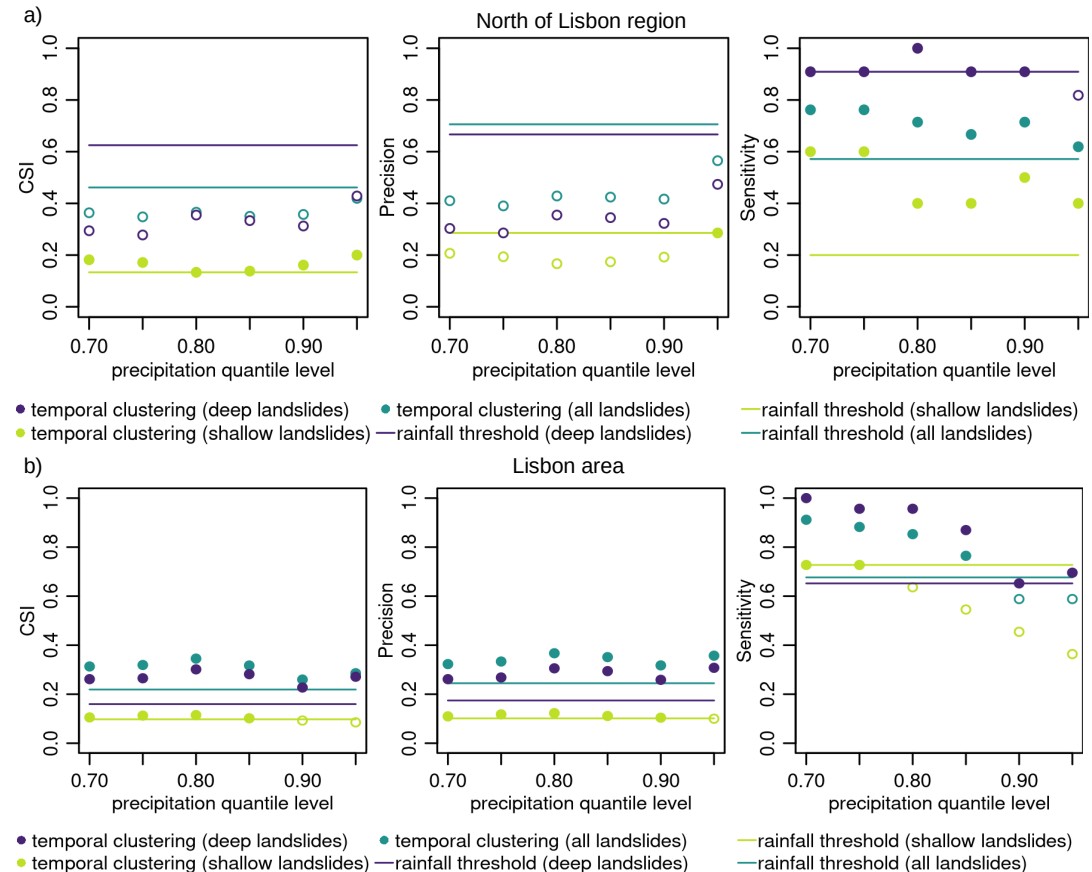

**Figure 10.** Cluster occurrence approach (dots) compared to the use of solely rainfall thresholds (lines) for a) North of Lisbon region and b) Lisbon area. The considered metrics for performance are: Critical Success Index (CSI); Precision (SR); Sensitivity (POD). Points are filled when the cluster-based approach outperforms the approach based solely on rainfall thresholds. Colors represent the subset of landslides used (see legend).

by the occurrence of three deep landslide events in both data sets. A further analysis that takes into account also the spatial dimension may provide further information about the interrelationship between landslide events.

The precipitation and meteorological conditions associated with landslides occurrence in Portugal, especially in the Lisbon region, have been studied in the past (e.g., Zêzere et al., 2005; Zêzere et al., 2015; Pereira et al., 2018). It was shown by Zêzere et al. (2005), that landslide occurrence can be associated with both the NAO index (North Atlantic Oscillation index) and the monthly rainfall anomalies, considering a 3-month moving average. Their analysis showed that months characterized by the occurrence of deep-seated landslides had very high values of average anomalous precipitation and intense negative average values of the NAO index. However, shallow landslide episodes, are not critically associated with NAO. In addition, Zêzere et al. (2015) showed the importance of the 30-day antecedent precipitation in the occurrence landslides in Portugal.

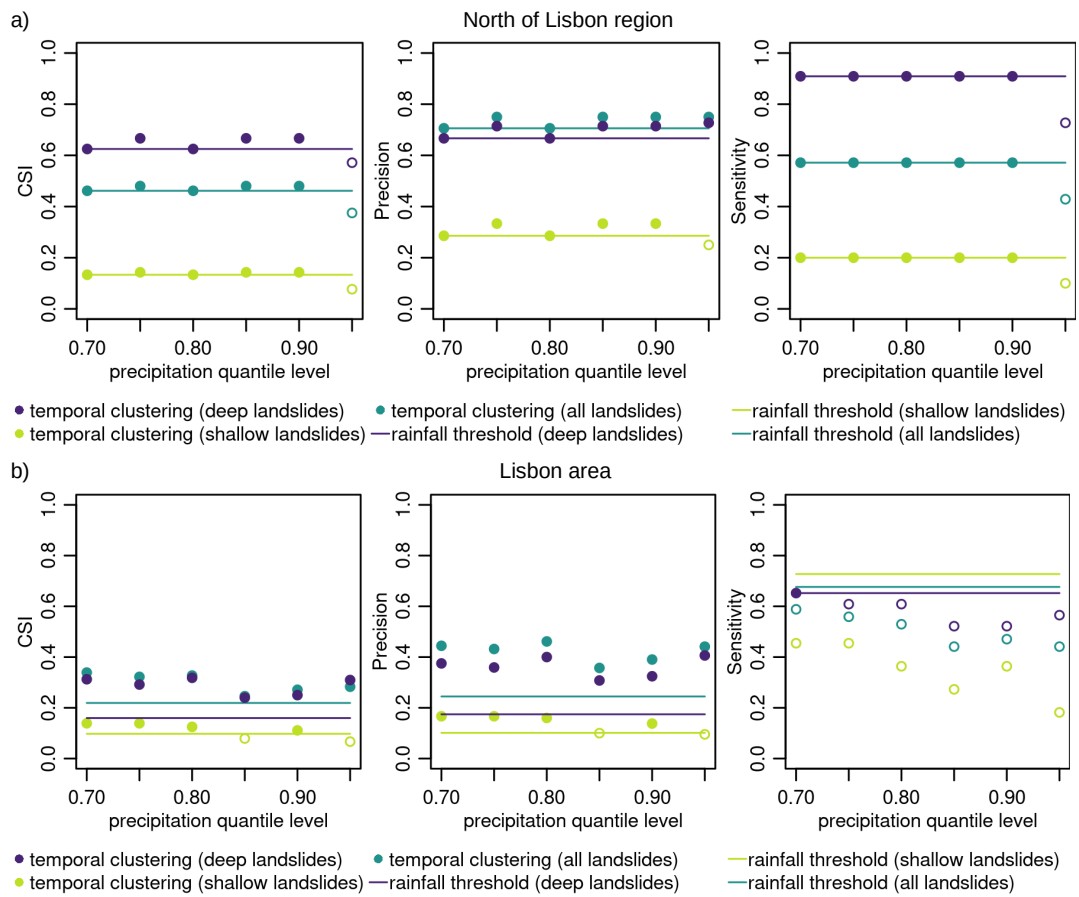

**Figure 11.** Cluster occurrence approach combined with rainfall thresholds (dots) compared to the use of solely rainfall thresholds (lines) for a) North of Lisbon region and b) Lisbon area. The considered metrics for performance are: Critical Success Index (CSI); Precision (SR); Sensitivity (POD). Points are filled when the cluster-based approach outperforms the approach based solely on rainfall thresholds. Colors represent the subset of landslides used (see legend).

This feature proves that landslides in mainland Portugal are usually anticipated by high rainfall in the 30-day period before, thus confirming the role of antecedent rainfall as a critical preparatory factor for landslide activity.

A closer look at the main meteorological conditions of the precipitation regimes and extreme precipitation in Portugal during winter shows a clear relationship with a few circulation weather types (Trigo and DaCamara, 2000). In particular, the Cyclonic (C) and those with a westerly component (SW/W/NW) contribute to a large percentage of the monthly precipitation. These westerly types are associated with the passage of frontal systems from west to east over the North Atlantic Basin (Cortesi et al., 2014; Ramos et al., 2014). Recent studies also show that over the western Iberian Peninsula extreme precipitation is often associated with the landfall of Atmospheric Rivers (ARs) (Ramos et al., 2014; Trigo et al., 2014; Eiras-Barca et al., 2016; Rebelo et al., 2018).

More recently, Pereira et al. (2018) analyzed a centennial catalogue of hydro-geomorphological events (including floods and landslides) and their atmospheric forcing. In accordance with the previous results, they concluded that the westerly flow and the cyclonic types are mainly associated with these hydro-geomorphological events. It was also shown that around 45% of the 130 events were somehow associated with the passage of an Atmospheric River. For comparison purposes, we extracted the circulation weather types, from the same database used by Pereira et al. (2018), for the trigger days of the landslides (62 days) analyzed in this paper. The results show that 75% of the landslides trigger days occurred during Cyclonic weather types and those with a westerly component, confirming the results previously found. An example is given for the hydrological year 2000/2001. Fig. 12 shows the daily rainfall (light vertical grey bars) from the IBERIA02 dataset along with the correspondent circulation weather type (in colors, Ramos et al. (2014)) for the hydrological year 2000/2001. In addition, if the occurrence of the ARs (Ramos et al., 2015) takes place during a particular day, this is highlighted in green in the daily rainfall bars. As mentioned before, some circulation weather types are associated with above average days or extreme rainfall events, specially the westerly circulation weather types like SW, N, NW and also the Cyclonic types (Ramos et al., 2014; Pereira et al., 2018).

Two landslide events occurred in January 2001 during the winter season (marked with red stars in Fig. 12). By the time of the landslides events occurrence, the accumulated precipitation since the beginning of the hydrological year (1st September) was above the 95th percentile of the hydrological year climatology. From the results in Fig. 8 we can see that the precipitation is associated with a temporal clustering of events. In particular, temporal clustering was observed over all the windows tested (up to 90 days) except for the smaller ones (below 7 days). The large-scale conditions that were associated with the anomalous accumulated precipitation over the hydrological year and specifically with the precipitation clustering can be easily assessed by analyzing the circulation weather types in Fig. 8 (in colors). Results show that from November 2020 onward, different periods of westerly flows (SW, W, NW, light blue) dominates the circulation over Portugal which contribute to a large extent to the anomalous accumulated precipitation in this water year. Extreme precipitation also occurred in December, a month prior to the landslide events, and those days were also associated with Atmospheric River landfall (precipitation bars highlighted in green) in accordance with Ramos et al. (2015).

The analysis of preconditioning meteorological variables have confirmed the influence of the Cyclonic (C) weather types and those with a westerly component (SW/W/NW) in the occurrence of landslides in Portugal, together with Atmospheric River landfall.

## 4.2   Landslides detection

The results in Fig. 10 and Fig. 11 are summarize in Table 2. They generally indicate that no method outperforms the other considering both datasets. The method solely based on temporal clustering of precipitation has better performances for Lisbon region both in terms of TP and FP, however, for North of Lisbon region it outperforms rainfall thresholds only looking at sensitivity. For regional management, if the interest is towards high sensitivity (number of correct positives as high as possible), the proposed approach based on precipitation cluster occurrence is a better choice than rainfall thresholds. On the contrary, the rainfall thresholds approach presents a higher precision (ratio of hits to the total number of events forecast). When considering the CSI, which includes both sensitivity and precision, there is not a method clearly outperforming the other one for both sites.

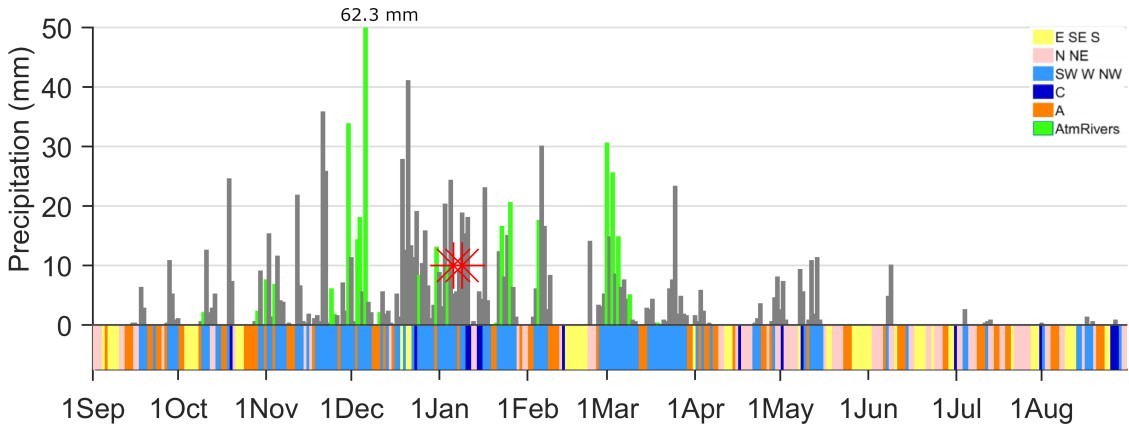

**Figure 12.** Daily rainfall (light vertical grey bars) along with the correspondent circulation weather type (in colours) for the first months of the 2000/2001 hydrological year. In addition, if the occurrence of the ARs takes place during this hydrological year is highlighted in green in the daily rainfall bars. The red stars indicate the landslides occurrence.

| Method | Sensitivity | Precision | Critical success Index | Area |
|---|---|---|---|---|
| Rainfall thresholds | 0.67 | 0.24 | 0.22 | Lisbon area |
| Temporal clustering (0.8) | 0.85 | 0.37 | 0.35 | Lisbon area |
| Combination (0.8) | 0.53 | 0.46 | 0.33 | Lisbon area |
| Rainfall thresholds | 0.57 | 0.71 | 0.46 | North of Lisbon region |
| Temporal clustering (0.8) | 0.71 | 0.43 | 0.37 | North of Lisbon region |
| Combination (0.8) | 0.57 | 0.71 | 0.46 | North of Lisbon region |

**Table 2.** Performance metrics of predictive methods. The performance of the temporal clustering method and its combination with the rainfall thresholds are reported for a quantile of 0.8

Regarding the combination on the temporal clustering information and rainfall threshold, we obtained similar performances for North of Lisbon region and better performances in terms of CSI and precision for Lisbon region. Then, if the interest is towards high precision (number of false as low as possible), the proposed approach based on precipitation cluster occurrence and rainfall threshold is a better choice than rainfall thresholds.

A cluster-based method depends on the choice of a quantile level to identify precipitation events. However, for a given index or region, we observed that the performances are consistent among most of the tested quantile levels, when compared to the rainfall threshold methods. Therefore, the dependence of the performance on the precipitation quantile is weak. Nevertheless, for the two areas analyzed we suggest a quantile level of 0.75, for overall better performances.

In general, for both methods, detecting shallow landslides is more difficult than deep ones. Fig. 6 showed the smaller dependence of shallow landslides to precipitation clusters. This might be due to the fact that a unique, heavy precipitation event may be enough to trigger shallow landslides (Corominas and Moya, 1999).

It is worth noticing that the proposed approach based on temporal clustering of rainfall does not include any information about rainfall totals, except in the initial selection of precipitation quantile. On the contrary, it retains information on the number of precipitation events preceding a landslide and therefore their closeness in time. The temporal dynamic of rainfall may therefore be important together with the cumulated volume over a window in the occurrence of landslides. The same precipitation amount falling in a single event rather than distributed over multiple events may result in higher total run-off and less infiltration. In addition to rainfall, also evaporation may influence the association between precipitation clusters and landslide occurrence, with higher evaporation weakening the precipitation-landslide statistical association. If strong evaporation occurs between two consecutive rainfall events belonging to the same cluster, then the effect of the first event is not seen in the second one. So it is like it occurred isolated.

## 4.3  Regional landslide events datasets

We observe differences in the method performances, depending on the dataset. This may be in part explained by the different nature of the inventories. For the Lisbon region, the landslides dataset was collected from newspaper sources, and a landslide event was considered to be an individual landslide or a set of landslides that occurred on a precise date (day) (Zêzere et al., 2014). In cases where different landslides occurred on consecutive days, each day was considered a landslide event and when the landslide activity was reported during several days, the first day of the period was assigned to the landslide event (Vaz et al., 2018). Such database is strongly dependent on consequences (landslides which caused damage to people: fatalities, injuries, missing and homeless people), hence in some cases the sign of the precipitation trigger was unclear. In general, only newsworthy content is reported by newspapers, which means that landslides that caused human damage or occurred in an urban environment are usually highlighted. For this reason, only landslides with a rainfall threshold with a return period of more than 3 years were used. The main aim was to reduce the possibility of including landslides with a triggering factor other than rainfall (e.g. human activity). Landslides with critical rainfall combinations with a return period of less than 3 years were assumed not to have been triggered by rainfall. On the contrary, rain-triggered landslides that did not cause social or economic damage were unlikely to have been reported in the newspapers (Zêzere et al., 2015; Vaz et al., 2018).

The second landslides dataset covers the region north of Lisbon. For this field-based landslides dataset, the regional criteria used to define a rainfall-triggered landslide is the identification of at least five individual landslides on natural slopes on a given date (Zêzere and Trigo, 2011; Zêzere et al., 2015). The temporal and spatial criteria for defining a rainfall-triggered landslide in the dataset north of Lisbon could be more difficult to apply regionally than the one associated to the single dates collected from the newspapers. The dataset distinguishes shallow (deep) landslide events as events for which more than 50% of landslide area is associated with slip surfaces below 1.5 m (above 1.5 m) (Zêzere and Trigo, 2011).

In fact in both datasets we can have both shallow and deep landslides occurring in the same landslide event. Nevertheless, that condition is more difficult to be verified in the Lisbon area dataset, associated with the Disaster database, mostly because it considers single dates and single landslide locations collected from newspapers.

The concept of a landslide event is not always straightforward. If we consider a landslide event associated with a single landslide, generally it is less difficult identifying the date of the event that will be then related to the daily rainfall data for the rainfall threshold assessment. (Vaz et al., 2018). This is in fact what the Disaster database, based on newspaper records, made possible for the Lisbon Area i.e. a very accurate definition of the temporal occurrence of the landslide events. However, the number of landslide events could be biased by the consequences criteria considered in the Disaster database. Zêzere et al. (2015) point out that the number of registered landslide events, in each study area in Portugal, is relatively low and this could make it difficult to establish reliable relationships between rainfall and landslides. In addition, several landslides could be triggered over consecutive days in a study area, and this may also be a potential source of uncertainty regarding the rainfall threshold definition, mostly due to the selection of the landslide event date, for which the rainfall threshold is defined Vaz et al. (2018).

We can conclude that the criteria to detect and collect landslides in the two datasets is not homogeneous and it has different problems and limitation, as detailed above. This results in a non homogeneous response data and in uncertainty in the statistical model. Ideally, landslide datasets with the same criteria could be a better benchmark and they could be used in further testing of the methods.

## 5 Conclusions

In the present work, we assess the effect of including information on the temporal clustering of rainfall for estimating the probability of landslide occurrence. We propose an empirical method, alternative or complementary to classical rainfall thresholds, that requires only precipitation data as an input. Our parsimonious approach proved to have a good power of detecting events compared with rainfall-threshold, with maximum CSI, considering all landslides, of about 0.4 in both cases. It has a higher sensitivity to landslide occurrence compared to classical rainfall thresholds, but lower precision, i.e. lower false negative (FN) but higher false positive (FP). One of the advantages of this method over rainfall thresholds is that it is not based on regression and does not include regimes information on rainfall totals. Although it requires the selection of a precipitation quantile to identify precipitation events, we observed a weak dependence on the quantile choice. Hence, it may prove a more general and transferable approach, overcoming the main limitation of absolute rainfall thresholds. This could be investigated by testing the method on other inventories of rainfall-triggered landslides in other countries. Different ways of combining the information about the temporal clustering of rainfall and rainfall totals could also be explored in future studies.

*Code and data availability.* Code and data available under request

*Author contributions.* CDM, FB, PR and EB conceived the methodology. SO took care of the data. FB took care of the analysis. JP and AR provided the meteorological interpretation of the results. FB wrote a first draft of the manuscript. All authors reviewed the manuscript.

*Competing interests.* At least one of the (co-)authors is a member of the editorial board of Natural Hazards and Earth System Sciences

*Acknowledgements.* S.C. Oliveira was financed by the project European ground motion risk assessment tool RASTOOL (Grant Agreement No. 101048474). E.B. has received funding from the European Union's Horizon 2020 research and innovation programme under grant agreement No 101003469 and from the DFG via the Emmy Noether Programme (grant ID 524780515). JGP thanks the AXA Research Fund for support. AMR is supported by the Helmholtz "Changing Earth" program and by the Portuguese Science Foundation (FCT) through the
project AMOTHEC (DRI/India/0098/2020). FB was financed by the RETURN project.

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
