# Peer review of "Temporal clustering of precipitation for detection of potential landslides"

_Natural Hazards and Earth System Sciences, 2023_

## Author Comment (AC1)

**Replies to referee #2**

**In section 2.2., time series window 1 to 90 days is considered based on what condition?**

The 90 days window is often used as the upper limit window size in the study of antecedent precipitation before landslides. Some example below:

Zêzere, J.L., Vaz, T., Pereira, S. et al. Rainfall thresholds for landslide activity in Portugal: a state of the art. Environ Earth Sci 73, 2917–2936 (2015). https://doi.org/10.1007/s12665-014-3672-0

Bevacqua, E., De Michele, C., Manning, C., Couasnon, A., Ribeiro, A. F. S., Ramos, A. M., et al. (2021). Guidelines for studying diverse types of compound weather and climate events. Earth's Future, 9, e2021EF002340. https://doi.org/10.1029/2021EF002340

In addition a good correlation of this time window with landslide activity was found in North Carolina by Fuhrmann et al (2008).

Christopher M. Fuhrmann , Charles E. Konrad & Lawrence E. Band (2008) Climatological Perspectives on the Rainfall Characteristics Associated with Landslides in Western North Carolina, Physical Geography, 29:4, 289-305, DOI: 10.2747/0272-3646.29.4.289

**In line 109, "(iv) the precipitation total preceding the landslide events, for windows of 1 to 90 days ending the day of the event, is computed".  Could this be explained?**

We hope the graphic representation may make it clearer. The idea is to cumulate precipitation falling before the landslide events. Each time we cumulate more precipitation up to 90 days of precipitation.

[Figure]

**Section 2.4. "we computed the presence of rainfall clustering preceding the second event with the modified series". How is the presence computed?**

We used the method explained in section 2.3 built upon a count-based procedure and a statistical test. The idea is that the number of precipitation events inside a window is Binomially distributed if there is no temporal clustering. The Binomial distribution is the discrete probability distribution of the number of successes in n independent trials. Each trial can have only two outcomes, yes or no, and the probability of having a yes in each individual trial is equal to p. The parameters of the distribution are therefore p and n. In this case each day is a trial and the outcome for each day is wet (yes) or dry (no). The probability of having a yes or no in a day is independent from the same probability in the other days, as an example, if day i is wet, it is not more probable that the following days are wet. This is true if the precipitation events are not clustered. If they are clustered, then, if day i is wet, it is more probable that the following days are wet. The assumptions of the binomial distribution are therefore not respected.

**Fig 9 legend symbols doesn't match the given plot**

Thanks for pointing this out. We will change it in the revised version.

**Section 4.2 Details about the connection of evaporation and precipitation**

**clusters?**

If a strong evaporation occurs between two consecutive rainfall events belonging to the same cluster, then the effect of the first event is not seen in the second one. So it is like it occurred isolated.

**Section 4.3. "In general, only newsworthy content is reported by newspapers, which means that landslides that caused human damage or occurred in an urban environment are usually highlighted. For this reason, only landslides with a rainfall threshold with a return period of more than 3 years were used. The main aim was to reduce the possibility of including landslides with a triggering factor other than rainfall (e.g. human activity). Landslides with critical rainfall combinations with a return period of less than 3 years were assumed not to have been triggered by rainfall"**

Thanks for the modification, we will introduce it into the text.

---

## Author Comment (AC2)

**Replies to Referee #1**

**GENERAL COMMENTS**

**1- English is generally good, but the manuscript would benefit from a careful and thorough revision. It seems that some parts (e.g. the discussion) were written in a rush, with typos and a flow of the text that can be improved.**

Thanks for the comment, we will improve the text, with a particular focus on the discussion section.

**2- The Methodology section misses a paragraph about the test site description. An appropriate test site figure is needed. Fig 1 could be used as well, but I suggest adding the meteorological stations used in the study and a clear subdivision showing which are the "Lisbon area" and the "North of Lisbon" area. Also, I suggest using two colors for deep and shallow landslides.**

We report below the new figure that will substitute Fig, 1 in the text. In the figure were not included the landslides associated to the north of Lisbon dataset. The figure with all those points will not be readable.  Regarding the site description we believe it is reasonably described in Section 2.1. However, it could be moved in a separate section called study area.

[Figure]

**3- While reading the manuscript, I got the feeling of reading a "regional" paper, which is a shame because the analysis and the conclusions are in my opinion are**

**interesting and relevant for a global readership. I therefore encourage to improve the introduction and the discussion, adding elements outside Portugal. Some good points could be e.g.: - to acknowledge that the subdivision between single landslide events and multiple (areal) landslide events is gaining consensus in a growing number of works (e.g. https://doi.org/10.1186/s40677-018-0105-5; https://doi.org/10.1016/j.geomorph.2017.07.032). - to discuss/introduce that the combined role of antecedent and peak precipitation is considered in recent works (https://doi.org/10.1007/s10346-023-02176-7; https://doi.org/10.1007/s10346-020-01505-4)**

Regarding the first point we will add in the Introduction the following consideration:

*An interesting characteristic of the two datasets is that they deal with areal and not single landslides. Older datasets, like the AVI dataset (Guzzetti et al., 1994), an inventory of landslides and floods that occurred in Italy, record landslides triggered by the same precipitation system as separate events. This is an important limitation when interested in identifying the landslides' triggers, therefore the subdivision between single landslide events and multiple landslide events is gaining consensus recently. For example, this is the case of the new catalog Frane Italia by Calvello and Pecoraro (2018), the LAND-deFeND database (Napolitano et al., 2018) or the work of Crozier (2017).*

Regarding the second point we will add in the Introduction in different points the following:

*In addition, recent works have considered in the investigation of landslides initiation the combined role of antecedent and peak precipitation (Kim et al., 2021; Nocentini et al., 2024). Antecedent rainfall, inducing a gradual increase in soil moisture and groundwater level, is an important factor in the determination of the slope stability and, therefore, the initiation of landslides (Rahardjo et al., 2001; Rahimi et al., 2011; Lee et al., 2012). Deep landslides, for example, may often be associated with monthly to seasonal fluctuations of the groundwater table. When the water table is high, light to moderate rainfall may provide sufficient water to trigger slope movement (Fuhrmann, Konrad and Band, 2008).*

*An innovative rainfall threshold is the one by Nocentini et al. (2024), which instead of considering thresholds based either on antecedent rainfall or peak intensity rainfall, propose a 3D threshold implementing both conditions.*

**4- The validation part needs to be supported by data. I suggest adding some tables to compare the metrics of the three approaches (e.g. at lines 225-229)**

We will add the following table in the text.

| Method | Sensitivity | Precision | Critical success Index | Area |
|---|---|---|---|---|
| Rainfall thresholds | 0.67 | 0.24 | 0.22 | Lisbon area |
| Temporal clustering (0.8) | **0.85** | 0.37 | **0.35** | Lisbon area |
| Combination (0.8) | 0.53 | **0.46** | 0.33 | Lisbon area |
| Rainfall thresholds | 0.57 | **0.71** | **0.46** | North of Lisbon region |
| Temporal clustering (0.8) | **0.71** | 0.43 | 0.37 | North of Lisbon region |
| Combination (0.8) | 0.57 | **0.71** | **0.46** | North of Lisbon region |

**5- I suggest placing figures just after they are mentioned in the text. Some of them are quite far away.**

We will try to modify this – note that using Latex we do not have full control on their positions. However, if necessary, we will remind the editorial board about this matter at the final stage before publication.

**SPECIFIC COMMENTS**

**L20 - this example about slope gradient could be removed. I think everyone reading NHESS is well aware of that.**

We will remove it

**L89-91 - the two datasets are very different. How this difference reflects into the results? Does it add uncertainty? That could be discussed in the discussion**

Part of this answer is already included in section 4.3 and it is related to the criteria associated with the definition of the landslide event, and therefore with the thresholds.

We will also include the following text in section 4.3

*The concept of a landslide event is not always straightforward. If we consider a landslide event associated with a single landslide, generally it is less difficult identifying the date of the event that will be related to the daily rainfall data for the rainfall threshold assessment. (Vaz et al., 2018). This is in fact what the Disaster database made possible for the Lisbon Area, based on newspaper records. A very accurate definition of the temporal occurrence of the landslide events. The number of landslide events could be biased by the consequences criteria considered in the Disaster database.*

*Zêzere et al. (2015), point out that the number of registered landslide events in each study area in Portugal, is relatively low and that could make it difficult to establish reliable relationships between rainfall and landslides. In addition, several landslides could be triggered over consecutive days in a study area, and this may also be a potential source of uncertainty regarding the rainfall threshold definition, mostly due to the selection of the landslide event date, for which the rainfall threshold is defined (Vaz et al., 2018).*

**L96- what if an event is composed by both deep and shallow slides?**

We will add a note at this point saying to see Section 4.3 for more information about the topic and we will improve Section 4.3 based on the considerations below:

In the case of the Lisbon region (Disaster database) were conceptually defined what is a Disaster event and a Disaster case. "A DISASTER case is a unique hydro-geomorphologic occurrence, which fulfills the DISASTER project database criteria, and is related to a unique space location and a specific period of time (i.e., the place where the flood or landslide harmful consequences occurred in a specific date). A DISASTER event is a set of DISASTER cases sharing the same trigger which can have a widespread spatial extension and a certain magnitude." Text extracted from Zêzere et al, 2014, DOI 10.1007/s11069-013-1018-y

In the North of Lisbon Region, a landslide event is any date for which at least five individual landslides are known to have occurred on natural slopes, considering the detailed data source. In addition, shallow (deep) landslide events were defined as being characterized by more than 50 % of landslide area associated to slip surfaces depth ≤ 1.5 m (depth ≥1.5 m) (Zêzere and Trigo 2011). Text extracted from Zêzere et al, 2015, DOI 10.1007/s12665-014-3672-0

In fact in both datasets we can have both shallow and deep landslides occurring in the same landslide event. Nevertheless, that condition is more difficult to verify in the Lisbon area dataset, associated with the Disaster database, mostly because it considers single dates and single landslide locations collected from newspapers. As far it was possible to observe that conditions do not occur in the Lisbon area dataset and were observed in the north of Lisbon dataset.

**L100- Please be clear on: what is the cell size (in meters) at this latitude.**

It is about 19.6 km. We will add it in the revised version of the manuscript.

**L100- How do you deal with multiple effects in which landslides are scattered across different cells?**

We selected a single grid-point that was nearest to the location of the landslides.

**Section 2.2 - Please state clearly if thresholds are already published (Zezere 2015) or if you updated them. Also, showing some equation or graph would be nice.**

We recomputed both the thresholds as specified at ll. 120. In particular, we performed a calibration and validation procedure. For each year, we calibrated the threshold using all years except the selected one. In this way we had a number of regression parameters equal to the number of years. The TP, TN, FN, and FP metrics in validation were computed for each of the excluded years.

Here is the graph of all regressions obtained for both of the region

[Figure]

And the equation with the average of the parameters confronted with the one computed in the paper.

Lisbon area:

$$y = 42.25 \cdot x^{0.6}$$

North of Lisbon:

$$y = 163.78 + 5.03 \cdot x$$

**L124-126 - this assumption requires a reference or an explanation**

We will add this in the text

*The Binomial distribution is the discrete probability distribution of the number of successes in n independent trials. Each trial can have only two outcomes, yes or no, and the probability of having a yes in each individual trial is equal to p. The parameters of the distribution are therefore p and n. In this case each day is a trial and the outcome for each day is wet (yes) or dry (no). The probability of having a yes or no in a day is independent from the same probability in the other days, as an example, if day i is wet, it is not more probable that the following days are wet. This is true if the precipitation events are not clustered. If they are*

*clustered, then, if day i is wet, it is more probable that the following days are wet. The assumptions of the binomial distribution are therefore not respected.*

**L171 - this is why the threshold should be visualized with a graph or an equation (see my previous comment)**

We reported the graph in previous comment, we will also add it in the text

**L181 forecast = forecasted**

We will change this.

**L186 - you do not consider TN in your metrics. I guess it is to avoid to have a distortion of the resulting values because of the numerical disproportion between TN and TP. Please, state it clearly in the text.**

We wanted to focus on extreme events detection. There are many "0" events but predicting the absence of events is not central here. We want to focus on landslides, their occurrence, how often we caught them or had a false alarm. We will state it clearly.

**L210 find = found**

We will change it.

**L233 -How do you combine? I guess by considering landslides only when the requirements of both methods are met... but this is not trivial, please state it clearly.**

Yes, we consider an alarm when both methods say that a landslide may occur. We will state it more clearly.

**L246-248 - This sentence is ot clear. Is this an example? Or 5 occurrences are considered "almost always"?**

This is an example. We added a : that was not necessary. The correct phrase should be: *Interestingly, five landslides occurred during the March-November period 1968–1969 in the Lisbon area, three deep landslide events, a shallow one, and again a deep one.*

Thanks for noticing it.

**L305-306 - Please, rephrase**

We will rephrase as follows:

*On the contrary, the rainfall thresholds approach presents a higher precision (ratio of hits to the total number of events forecast). When considering the CSI, which includes both sensitivity and precision, there is not a method clearly outperforming the other one for both sites.*

**L327 - dataset of was = dataset was**

We will change this.

**Section 4.3 - You discuss these differences, but you should also mention if and how they affect the results. Moreover, if you consider it a limitation, you should mention it. (I think it is nice when papers address also the main limitations of the works)**

1. If and how they affect the results: the criteria to detect landslides are not the same, therefore the response data is not homogenous (whereas the drivers data is, same precipitation dataset). This is bringing uncertainty in the statistical model.
2. It is a limitation. Ideally, 2 landslide datasets with the same criteria would be better.